# How many begomovirus copies are acquired and inoculated by its vector, whitefly (*Bemisia tabaci*) during feeding?

**Buddhadeb Roy, Prosenjit Chakraborty, Amalendu Ghosh** *

Advanced Centre for Plant Virology, ICAR-Indian Agricultural Research Institute, New Delhi, India

* amal4ento@gmail.com

**Data Availability Statement:** The sequences generated in this study are available in the NCBI GenBank (accession numbers: MT920041, MW399221, and MW399222).

## Abstract

Begomoviruses are transmitted by whitefly (*Bemisia tabaci* Gennadius, Hemiptera: Aleyrodidae) in a persistent-circulative way. Once *B. tabaci* becomes viruliferous, it remains so throughout its life span. Not much is known about the copies of begomoviruses ingested and/or released by *B. tabaci* during the process of feeding. The present study reports the absolute quantification of two different begomoviruses viz. tomato leaf curl New Delhi virus (ToLCNDV, bipartite) and chilli leaf curl virus (ChiLCV, monopartite) at different exposure of active acquisition and inoculation feeding using a detached leaf assay. A million copies of both the begomoviruses were acquired by a single *B. tabaci* with only 5 min of active feeding and virus copy number increased in a logarithmic model with feeding exposure. Whereas, a single *B. tabaci* could inoculate 8.21E+09 and 4.19E+11 copies of ToLCNDV and ChiLCV, respectively in detached leaves by 5 min of active feeding. Virus copies in inoculated leaves increased with an increase in feeding duration. Comparative dynamics of these two begomoviruses indicated that *B. tabaci* adult acquired around 14-fold higher copies of ChiLCV than ToLCNDV 24 hrs post feeding. Whereas, the rate of inoculation of ToLCNDV by individual *B. tabaci* was significantly higher than ChiLCV. The study provides a better understanding of begomovirus acquisition and inoculation dynamics by individual *B. tabaci* and would facilitate research on virus-vector epidemiology and screening host resistance.

## Introduction

The whitefly, *Bemisia tabaci* (Gennadius, Hemiptera: Aleyrodidae) is one of the world's top 100 invasive species (IUCN, http://www.iucngisd.org/gisd/100_worst.php), which causes severe losses to more than 900 plant species [1, 2]. The considerable genetic diversity and varying abilities to interbreed have led to the belief that *B. tabaci* represents a cryptic species complex [3–6]. To date, 45 such cryptic species of *B. tabaci* are known [7]. Besides direct damage caused by feeding, *B. tabaci* transmits begomoviruses, carlaviruses, closteroviruses, criniviruses, ipomoviruses, nepoviruses, potyviruses, torradoviruses, and a rod-shaped DNA virus [8–10]. Begomoviruses, in general, are transmitted by *B. tabaci* in a persistent-circulative manner. To date, 445 species of begomoviruses are known (ICTV, 2020) [11]. Begomoviruses are

**Funding:** The author(s) received no specific funding for this work.

**Competing interests:** The authors have declared that no competing interests exist.

single-stranded (ss) circular DNA viruses belong to the family *Geminiviridae*. Infected plants show yellow mosaic, vein yellowing, leaf curling, stunting, and vein thickening. Yield losses of 20–100% in economic crops are reported due to *B. tabaci* and begomoviruses [12]. Most of the begomoviruses are phloem-limited and are restricted to the vascular system except bean dwarf mosaic virus that invades mesophyll tissue [13]. Mouthparts of *B. tabaci* are modified to piercing the leaf tissue and reach the vascular system to suck plant sap. The maxillary stylets are held opposed and form the food and salivary canal. *B. tabaci* inserts its stylets in the vascular system of infected plants and ingests begomovirus particles. Transmission of tomato yellow leaf curl virus (TYLCV) by *B. tabaci* is widely studied [14]. TYLCV virions pass along the food canal and reach the digestive tract after 40 min of ingestion [15]. Once the virions enter the alimentary system of *B. tabaci*, it is received by receptors located at midgut of *B. tabaci*. It is thought that the virus crosses the filter chamber and the midgut barriers to enter the hemolymph after 90 min [15]. The virions are circulated through hemolymph in coated vesicles and translocated into the primary salivary glands where they can be detected 4–7 h after the onset of feeding [15]. The virions are egested with the saliva into the plant phloem during feeding and salivation. Several studies reported the diagnostics, epidemiology, host plant resistance, and transmission of *B. tabaci*-borne begomoviruses. Not much is studied about the copies of begomoviruses that are ingested during sucking phloem sap or egested with saliva by its vector, *B. tabaci*. The amount of TYLCV-DNA accumulated in its vector, *B. tabaci* was earlier estimated by quantitative chemiluminescent dot-blot assay [16]. Watermelon chlorotic stunt virus (WmCSV) and TYLCV-DNA have been quantified from group of poor and efficient transmitter strain of *B. tabaci* [17]. None of these studies quantified the virus dynamics in individual *B. tabaci*. The present study measures the uptake and release of one bipartite, tomato leaf curl New Delhi virus (ToLCNDV) and one monopartite, chilli leaf curl virus (ChiLCV) begomoviruses by individual *B. tabaci* using a detached leaf assay. ToLCNDV contains two similar-sized genomic components, DNA-A and DNA-B, each approximately 2.7 kb in size [18, 19]. ToLCNDV is responsible for significant yield losses in several crops under the Solanaceae and Cucurbitaceae families [20–22]. ChiLCV is monopartite and approximately 2.7 kb in size, has emerged as a serious threat to chilli production [23, 24]. The number of begomovirus copies acquired and transmitted by individual *B. tabaci* is important to shed light on the efficiency of virus transmission and useful in understanding the disease epidemiology.

## Materials and methods

### Insect vector

The initial population of *B. tabaci* was collected from a stock culture maintained at Advanced Centre for Plant Virology, Indian Agricultural Research Institute (IARI), New Delhi. A single adult female *B. tabaci* was released on eggplant (var. Navkiran, Mahyco) to establish a homogeneous population. The identity of the homogenous population was characterized by morphological keys [25] and sequencing mitochondrial cytochrome oxidase subunit I (mtCOI). The population was maintained under controlled environmental conditions at 28 ± 2° C, 60 ± 10% relative humidity, and 16 hrs light-8 hrs dark photoperiod. A representative sample of the population was tested in PCR to confirm its aviruliferous status. The freshly emerged adult females were collected using an aspirator for further experiments.

### Host plant

Eggplant (var. Navkiran), tomato (var. S-22, Icon Seeds), and chilli (var. HPH-1041, Syngenta) were grown in plastic pots filled with soil rite mixture. The plants were maintained in a growth chamber under insect-proof conditions. Eggplant was used to maintain the homogenous

population of *B. tabaci*. Tomato and chilli were used to maintain the pure culture of begomoviruses, acquisition, and inoculation by *B. tabaci*.

## Virus isolates

The inoculums of ToLCNDV and ChiLCV have been collected from pure cultures maintained at Division of Plant Pathology, IARI, New Delhi. ChiLCV was maintained in chilli (var. HPH-1041) and ToLCNDV was maintained in tomato (var. S-22) plants by *B. tabaci*-inoculation. The identity of the begomoviruses was confirmed by PCR as described below. Both the begomoviruses were maintained under insect-proof conditions for further experiments.

## DNA extraction from *B. tabaci* and host plants

Total DNA was extracted from *B. tabaci* adults, tomato, and chilli leaves using cetyltrimethylammonium bromide (CTAB) methods [26]. CTAB extraction buffer was prepared in a total volume of 1 ml containing 100 mM Tris-HCl pH 8.0, 1.4 M NaCl, 20 mM EDTA pH 8.0, 2% CTAB, and 2 µl β-mercaptoethanol. *B. tabaci* specimens were crushed in 100 µl of CTAB extraction buffer inside a microfuge tube using a homogenizer. The tomato and chilli leaves were crushed in 1 ml of extraction buffer using mortar pestles. The lysate was vortexed and incubated at 65˚C for 30 min. An equal volume of chloroform: isoamyl alcohol (24:1) was added to the lysate and centrifuged at 16,000 x g for 15 min. The upper aqueous phase was transferred to a fresh microfuge tube. The DNA was precipitated by adding 0.7 volume of ice-cold isopropanol and kept at -20˚C for 1 hr. The sample was centrifuged at 16,000 x g for 15 min and the supernatant was decanted gently. The pellet was gently washed with 100 µl of 70% ethanol. The ethanol was decanted and residual ethanol was removed by drying at room temperature. The pellet if any was dissolved in 20 µl sterile distilled water in the case of *B. tabaci* and 50 µl in the case of a leaf sample. The concentration and purity of DNA samples were determined in a spectrophotometer (NanoDrop One, Thermo Scientific, USA).

## Characterization of *B. tabaci* and begomoviruses

*B. tabaci* was randomly collected from the homogeneous population and DNA was extracted as described above. A partial fragment of the mtCOI gene was amplified using primer pairs C1-J-2195 5′–TTGATTTTTTGGTCATCCAGAAGT–3′ and TL2-N-3014 5′–TCCAATGCAC–TAATCTGCCATATTA–3′ [27]. Begomovirus infection was tested by PCR with universal degenerate primers for begomoviruses (Begomo F: 5′–ACGCGTGCCGTGCTGCTGCCCC–CATTGTCC–3′ and Begomo R: 5′–ACGCGTATGGGCTGYCGAAGTTSAGAC–3′) targeting DNA-A [28]. PCR was carried out in a 25 µl reaction mixture containing 1x reaction buffer, 4.0 mM MgCl$_2$, 0.4 mM dNTPs, 0.625 U *Taq* DNA polymerase, 10 pmole of each forward and reverse primer, and DNA template (~50 ng for *B. tabaci*, ~100 ng for plants). The PCR was performed in a T100 Thermal Cycler (Bio-Rad). The reaction mixture with C1-J-2195 and TL2-N-3014 primers followed one cycle of initial denaturation at 94˚C for 5 min followed by 35 cycles of 94˚C for 30 s, 53˚C for 30 s, 72˚C for 1 min and a final extension at 72˚C for 10 min. The PCR with Begomo F and Begomo R primers followed one cycle of initial denaturation at 94˚C for 5 min followed by 35 cycles of 94˚C for 30 s, 57˚C for 45 s, 72˚C for 1 min 30 s and a final extension of 10 min at 72˚C. The PCR products were resolved on 1% agarose gel stained with GoodView (BR Biochem, India) and visualized in a gel documentation system (MasteroGen Inc, Taiwan) with 100 bp plus DNA ladder (Thermo Scientific). Purified PCR products were ligated to pGEM-T Easy Vector (Promega, UK) following the manufacturer's protocol and transformed into DH5α *E. coli* cells [29]. Plasmid DNA was extracted using Wizard Plus SV Minipreps DNA Purification System (Promega) and sequenced. The sequences

were processed by BioEdit and BLASTn was performed to check the species homology. Consensus sequences were submitted in the GenBank. As *B. tabaci* is considered as a complex of cryptic species, a Bayesian Inference phylogeny was undertaken using MrBayes 3.2 considering genetic divergence cutoff of 4% [30].

### Acquisition and inoculation of begomoviruses by *B. tabaci*

Young *B. tabaci* adult females with a maximum age of 24 hrs were used for virus acquisition and inoculation. The aviruliferous adult females were collected using an aspirator. Insect breeding dishes (50 mm d, 15 mm h) with a mesh on lid were used for acquisition and inoculation of virus by *B. tabaci*. 5 ml of 0.9% agar-agar was poured in the bottom half of the insect breeding dish at about 50˚ tilt (Fig 1). Petioles of the apical symptomatic leaves of ToLCNDV and ChiLCV-infected 45-day old plants were inserted in the solidified agar-agar and used for virus acquisition by individual *B. tabaci*. The virus copies in source plants were also estimated. In case of inoculation, leaf of 45-day old virus-free tomato and chilli plants was used for virus inoculation by *B. tabaci*. Leaves of same size and age (approximately 6.2 square cm) were considered throughout the experiment. A single aviruliferous *B. tabaci* was released on the virus-infected leaf for acquisition of the virus. For virus inoculation, individual *B. tabaci* was released on ChiLCV- and ToLCNDV-infected leaves separately for 24 hrs of virus acquisition and shifted to virus-free leaves inside the insect breeding dishes. *B. tabaci* within the dishes were constantly monitored initially under a stereomicroscope to confirm if they started feeding on leaves. All the acquisition and inoculation sets were kept in dark at 28 ±2˚C, and 60 ±10% relative humidity. *B. tabaci* adults were collected from acquisition and inoculation setups at 1 min, 5 min, 10 min, 20 min, 1 hr, 2 hrs, 6 hrs, 12 hrs, and 24 hrs post release. The setups where adults settled on leaf and started feeding, were only considered for quantification of virus titre. Separate sets were maintained for ToLCNDV and ChiLCV and replicated thrice.

### Standardization of real-time PCR

Three pairs of real-time PCR primers each for ToLCNDV and ChiLCV were designed based on the coat protein (CP) sequences of the viruses (Table 1). All CP sequences of ToLCNDV

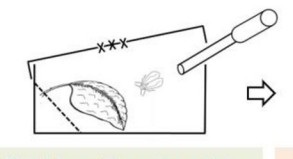
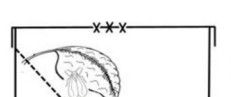
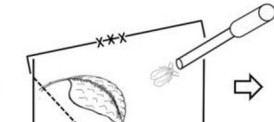
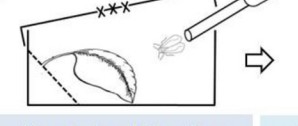
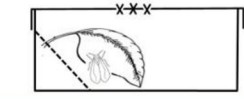

**Fig 1. Ingestion and egestion of ToLCNDV and ChiLCV by single *B. tabaci* using detached leaf assay.**

**Table 1. List of primer pairs used in the study.**

| Forward primer | | | Reverse primer | | | Amplicon size (bp) | Target region | Reference |
|---|---|---|---|---|---|---|---|---|
| Primer name | Primer sequence (5'-3') | Melting temperature (°C) | Primer name | Primer sequence (5'-3') | Melting temperature (°C) | | | |
| C1-J-2195 | TTGATTTTTTGGTCATCCAGAAGT | 52 | TL2-N-3014 | CCAATGCACTAATCTGCCATATTA | **52** | 860 | *B. tabaci* mtCOI | [27] |
| Begomo F: | ACGCGTGCCGTGCTGCTGCCCCCATTGTCC | 71.2 | Begomo R: | ACGCGTATGGGCTGYCGAAGTTSAGAC | **63.1** | 2800 | Begomovirus DNA A | [28] |
| AG149F | TGAACAGGCCCATGAACAG | 55.3 | AG150R | ACGGACAAGGAAAAACATCAC | 53.5 | 290 | ChiLCV coat protein | This study |
| | | | AG152R | CGGACAAGGAAAAACATCACAC | 54.6 | 290 | | |
| AG153F | ACAGGCCCATGAACAGGAAG | 55.5 | AG154R | CACGGACAAGGAAAAACATCAC | 54.6 | 280 | | |
| AG155F | GCCGATGAACAGAAAACCC | 54.2 | AG156R | TTTCACACAGAATCGCTTCC | 53.1 | 200 | ToLCNDV coat protein | |
| | | | AG158R | TCACACAGAATCGCTTCCC | 54.9 | 190 | | |
| | | | AG160R | TCGTGCCGAGATTCAAAAG | 53.0 | 100 | | |

and ChiLCV available in the NCBI were utilized for primer designing using software Primer 3 Input version 0.4.0. The major aspects such as primer length, amplicon size, GC contents, and intra-primer or inter-primer homology were taken into consideration while designing the primers. The site-specificity of the primers was verified by performing Primer-BLAST (www.ncbi.nlm.nih.gov). Conventional PCR was carried out in a 25 μl reaction mixture to validate the newly designed primers. Each reaction contained 1x reaction buffer, 4.0 mM MgCl₂, 0.4 mM dNTPs, 0.625 U *Taq* DNA polymerase, 10 pmole of each forward and reverse primer, and DNA template (~50 ng for *B. tabaci* and ~100 ng for plants), and 8.5 μl of nuclease-free water. The PCR was performed in T100 Thermal Cycler (Bio-Rad) with one cycle of initial denaturation at 95°C for 5 min, 35 cycles of denaturation at 95°C for 30 s, annealing at temperature gradient 50–54°C for 30 s based on the melting temperature of the primer sets, and extension at 72°C for 40 s followed by a final extension at 72°C for 10 min. The PCR products were resolved on 2% agarose gel stained with GoodView and visualized in a gel documentation system with 100 bp plus DNA ladder. One primer pair each for ToLCNDV and ChilCV with the best amplification was considered for real-time PCR.

Real-time PCR was performed in a 48-well Insta Q48m (Himedia, India) with 20 μl reaction volume consisted of a 10 μl 2x DyNAmo ColorFlash SYBR Green qPCR Mix (Thermo Scientific), 2 μl template DNA (~100 ng for plant and ~50 ng for *B. tabaci*), and 10 pmole of each forward and reverse primer. Thermal cycling was performed as initial denaturation at 95°C for 5 min, 30 cycles of 95°C for 30 s, 53°C for 30 s, and 72°C for 40 s. Since SYBR Green I dye binds non-specifically to any double-stranded DNA, dissociation or melting stage was carried out after every reaction to determine the specificity of the amplicons based on the melting curve. Each of the biological replicates had three technical replicates. A no-template control (NTC) and positive control were included in each assay. The $C_T$ values thus obtained were used to calculate the mean $C_T$ and standard error of mean (SEM) using the Microsoft Excel software.

## Preparation of standard curve

For preparing a standard curve of ToLCNDV and ChiLCV, the PCR amplified product of ToLCNDV and ChiLCV using primer pairs AG155F-AG158R and AG149F-AG150R, respectively were ligated in pJET1.2 vector using CloneJET PCR Cloning Kit (Thermo Scientific) and transformed into DH5α *E. coli* cells. A ten-fold serial dilution of the linearized plasmid DNA from $5 \times 10^2$- $5 \times 10^{-7}$ ng was subjected to real-time PCR. The real-time PCR was carried

out in 20 μl reaction volume as described above taking serially diluted plasmid DNA as templates. To determine the assay reproducibility each dilution of the plasmid DNA used for constructing the standard curve of ToLCNDV and ChiLCV was replicated thrice. The standard curves for ToLCNDV and ChiLCV were prepared by plotting a linear regression curve with the mean $C_T$ values on Y-axis and log DNA dilution in ng on X-axis.

### Quantification of begomovirus copies

To quantify the virus copies, ToLCNDV and ChiLCV-exposed *B. tabaci* and inoculated leaves were collected at different time exposure and DNA was extracted as described above. The absolute quantification of ToLCNDV and ChiLCV DNA ingested and egested by individual *B. tabaci* was carried out by fitting the mean $C_T$ values into corresponding standard curves. The virus titre calculated in ng by extrapolating the mean $C_T$ values in the standard curve was converted into the copy number using the formula N = (X ng * 6.0221x $10^{23}$ molecules/ mole)/ (n * 660 g/ mole * 1x$10^9$ ng/g), where N is the number of viral copies; X is the amount of amplicon in ng, and n is base pairs of recombinant plasmids. Based on the total volume of extracted DNA, the average copy numbers of ToLCNDV and ChiLCV in individual *B. tabaci* and inoculated leaves at different periods of feeding were calculated along with SEM. The log of copy number of viruses ingested and egested by individual *B. tabaci* was modeled using logarithmic function separately. A partial F-test was performed to test the difference between the rate of increase in copy number of ToLCNDV and ChiLCV.

## Results

### Characterization of *B. tabaci*

The homogeneous population of *B. tabaci* was characterized by both morphological features and sequencing. The adult *B. tabaci* had a yellowish body, red compound eyes, and two pairs of white wings that were covered with powdery wax. The wings were in "inverted V" orientation at rest. Initially the *B. tabaci* nymphs were translucent that became yellow to pale green at later stage of life cycle. The fourth instar nymph exhibited 7 pairs of setae at dorsum. PCR amplification of *B. tabaci*-mtCOI gene with C1-J-2195 and TL2-N-3014 primers showed an expected amplicon of ~860 bp on agarose gel. BLASTn analysis showed 99.99% homology with other *B. tabaci* sequences in NCBI. The sequence can be retrieved using the GenBank Accession No. MT920041. Bayesian Inference phylogeny considering genetic divergence cut-off of 4% revealed that the population belonged to cryptic species *B. tabaci* Asia II 1 (data not presented).

### Characterization of the begomoviruses

PCR with ToLCNDV and ChiLCV-inoculated samples produced ~2.7 kb products. The bidirectional sequencing of the products could generate 1913 nt (GenBank Accession No. MW399221) and 1896 nt (GenBank Accession No. MW399222) sequences for ToLCNDV and ChiLCV, respectively. In BLASTn analysis, both the sequences showed 100% similarity with other ToLCNDV and ChiLCV DNA-A sequences available at NCBI.

### Standardization of real-time PCR assay

All the six primer pairs showed a single specific amplicon without any cross-reactivity in both *in silico* and *in vitro* PCR analysis. In gradient PCR, all the primer pairs produced specific amplicon at 50 to 55°C annealing temperature. Based on the amplification in conventional PCR at annealing temperature of 53°C, primer pair, AG155F-AG158R for ToLCNDV and

AG149F-AG150R for ChiLCV were selected for real-time PCR assay. Primer pair, AG155F-AG158R and AG149F-AG150R produced prominent bands of 190 and 290 bp, respectively. Both the primer pairs did not produce a secondary peak in the melting curve analysis in real-time PCR (S1 Fig). The specific melting temperature for both ToLCNDV and ChiLCV products was around 81˚C. The standard curve of ToLCNDV and ChiLCV showed high amplification efficiency of 103.72% and 112.29% respectively indicating optimal conditions for absolute quantification. The standard curves showed coefficient of correlation ($R^2$) of 0.9827 and 0.9950, respectively, and covered a linear range from $10^2$–$10^9$ copies of viral DNA.

## Quantification of begomovirus acquired by *B. tabaci*

The virus copies in the source leaves used for acquisition of ToLCNDV and ChiLCV by *B. tabaci* were 3.34E+17 and 2.94E+17, respectively. No viral load could be detected in individual *B. tabaci* up to 1 min of active acquisition feeding for both the begomoviruses. ToLCNDV and ChiLCV were detectable post 5 min of active feeding by *B. tabaci*. ToLCNDV and ChiLCV copy numbers ingested by single *B. tabaci* were 4.10E+09 and 2.10E+11, respectively post 5 min of feeding (S1 Table). The virus copies increased in *B. tabaci* with increased feeding exposure. A steep increase of ToLCNDV load in individual *B. tabaci* was observed 6 hrs onwards. ToLCNDV copy number in individual *B. tabaci* was 2.05E+12 at 6 hrs and reached a peak of 4.64E+14 copies 24 hrs post feeding. The rate of increase of ToLCNDV copies was fitted in a logarithmic model ($R^2 = 0.75$) (Fig 2). Whereas, in case of ChiLCV, the virus copies started accumulating sharply post 2 hrs of acquisition feeding. The increase in ChiLCV copies in individual *B. tabaci* followed a logarithmic model ($R^2 = 0.68$). The ChiLCV copies in individual *B. tabaci* were significantly higher than ToLCNDV throughout the feeding period. The ChiLCV copy numbers increased up to 6.40E+15 which was 14-fold higher than ToLCNDV at 24 hrs of feeding. The rate of virus ingestion by individual *B. tabaci* was significantly higher (partial F-test, p<0.001) in case of ChiLCV than ToLCNDV.

## Quantification of begomovirus egested by *B. tabaci*

The amount of virus egested by individual *B. tabaci* was quantified in detached leaves. None of the begomoviruses egested by a single *B. tabaci* could be detected at 1 min of active feeding. However, both the begomoviruses could be detected in real-time PCR 5 min onwards. ToLCNDV and ChiLCV copy numbers egested by individual *B. tabaci* in detached tomato and chilli leaves were 2.20E+10 and 2.10E+09, respectively 5 min post feeding (S1 Table). The virus copy number increased in detached leaves with increased feeding exposure of *B. tabaci*. The virus copies increased to 5.52E+12 and 6.02E+11, respectively for ToLCNDV and ChiLCV 2 hrs post inoculation feeding. *B. tabaci* released around 10-fold higher copies of ToLCNDV than ChiLCV up to 6 hrs of feeding. Copy number of both the viruses started increasing steeply post 6 hrs of feeding. ToLCNDV copy number was 4.19E+13 at 6 hrs and reached up to 9.80E+15 24 hrs post feeding. Whereas, individual *B. tabaci* egested 3.38E+12 copies of ChiLCV at 6 hrs and increased up to 8.31E+13 copies 24 hrs post feeding on detached leaf. The rate of increase in virus copies egested by individual *B. tabaci* followed a logarithmic model for both the viruses ($R^2 = 0.78, 0.75$) (Fig 2). However, the rate of increase of ToLCNDV was significantly higher (partial F-test, p<0.001) than ChiLCV.

## Discussion

*B. tabaci* vectors large number of plant viruses of the genera *Begomovirus*, *Crinivirus*, *Carlavirus*, *Closterovirus*, and *Ipomovirus* [31, 32]. The genetic material, shape of the virion, and the mode of transmission of these viruses are very different. Begomoviruses are small circular

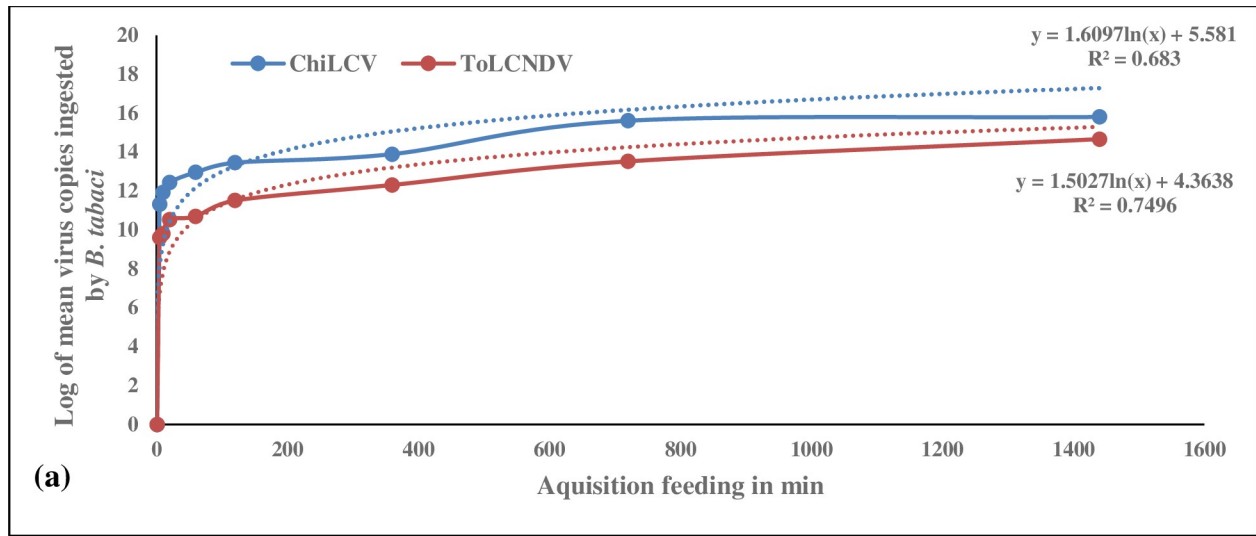

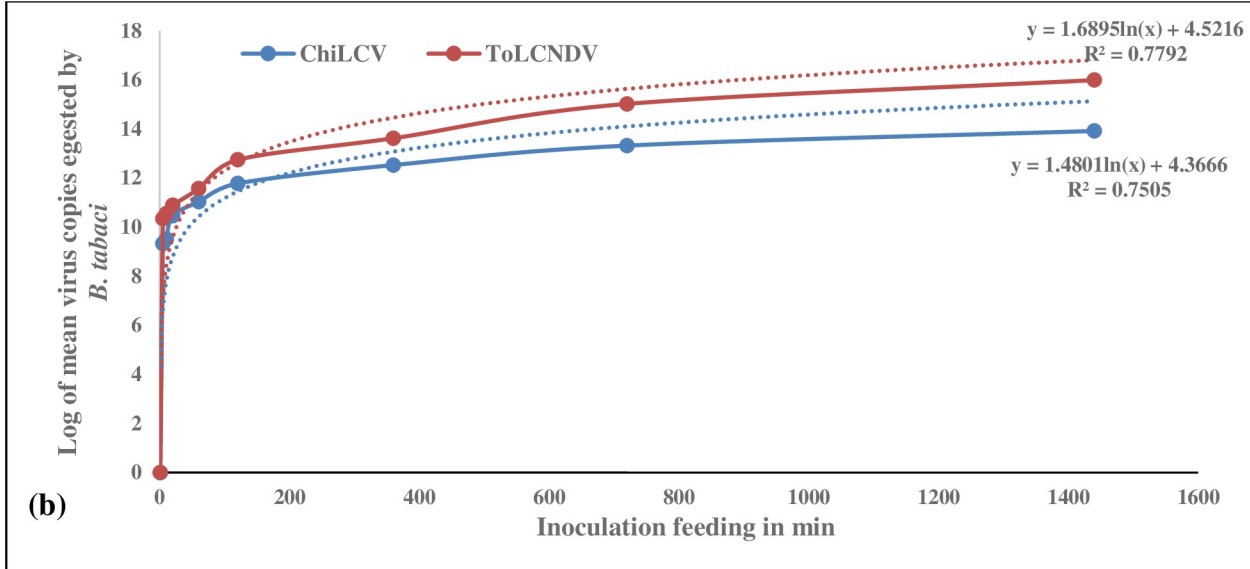

**Fig 2.** Dynamics of ToLCNDV and ChiLCV copies ingested (a) and egested (b) by individual *B. tabaci*. Feeding exposure is expressed in min on X-axis and log of virus copy number is plotted on Y-axis. ToLCNDV and ChiLCV copies were estimated at 1 min, 5 min, 10 min, 20 min, 1 hr, 2 hrs, 6 hrs, 12 hrs, and 24 hrs of feeding. The increase in virus copies is marked by maroon (ToLCNDV) and blue (ChiLCV) lines. The dotted lines indicate a best-fit logarithmic curved line to illustrate predicted increase of virus copies with time. $R^2$ values and equations are mentioned on the graph.

ssDNA viruses transmitted by *B. tabaci* in a persistent circulative way. *B. tabaci* ingests the begomovirus particles during feeding on phloem sap of infected plants. Virions pass through the food canal and enter the digestive tract. Midgut, especially the filter chamber is the site for begomovirus translocation. Virions are transported through the cytoplasm of filter chamber in vesicles that fuse with basal plasma membrane and released in the hemocoel. The interaction of begomovirus with *B. tabaci* has been the subject of many studies for the last fifty years [33]. During the entire process, the viral proteins need to interact with several midgut and primary salivary gland proteins [34]. Coat protein (CP) of the begomovirus interacts with the various insect tissues to ensure further translocation within *B. tabaci* until egestion. Several proteins in the midgut are known to bind with the begomovirus CP and influence the transmission. Heat

shock protein 70, peptidyl-prolyl isomerases protein, peptidoglycan recognition protein genes regulate begomovirus transmission by *B. tabaci* [35–37]. CP of TYLCV interacts with a 63-kDa GroEL chaperone produced by endosymbiotic bacteria to protect the virus in hemolymph [38, 39]. The virions gradually accumulate in the primary salivary gland through receptor-like elements and egested with saliva during feeding.

The involvement of several receptors in interactions of *B. tabaci* and begomoviruses alters the transmission competence of different *B. tabaci* cryptic species and virus species combinations [34, 40–42]. *B. tabaci* MEAM1 was found to be a better transmitter of bhendi yellow vein mosaic virus in comparison to *B. tabaci* Asia 1 [43]. Cotton leaf curl Multan virus was reported to be successfully transmitted by *B. tabaci* Asia II 7, but MEAM1 and MED were unable to transmit the same [44]. The present study reports the absolute quantification of ToLCNDV and ChiLCV copies ingested and egested by *B. tabaci* cryptic species Asia II 1. *B. tabaci* Asia II 1 could acquire $>10^9$ copies of ToLCNDV and ChiLCV within 5 min of feeding. ChiLCV copies acquired by an individual *B. tabaci* were 51-fold higher than ToLCNDV at 5 min of active feeding. Previously it was reported that ChiLCV needs at least 10 min to 3 hrs of acquisition access period (AAP) for successful transmission [45]. However, the results of the present study indicate that *B. tabaci* could acquire millions of copies of ChiLCV by only 5 min of active feeding. The ingestion of virus copies increased with increase in feeding exposure. The ChiLCV copy number started increasing sharply post 2 hrs of feeding. The rate of increase followed a logarithmic model. Whereas, maximum increase of ToLCNDV copies in individual *B. tabaci* was 6 hrs post feeding. The difference may be due to the efficiency of the virus CP to interact with *B. tabaci* receptors to become circulative. However, Kollenberg et al. [17] suggest variation in virus uptake may be due to the variation in virus concentration in source plants. To overcome the variation in viral DNA in source plant, in the present study, the top leaf of plants of same physiological age were used. The viral copies in the source plants of ToLCNDV and ChiLCV were almost equal. No virus copy was detected in *B. tabaci* by real-time PCR immediately (1 min) after the virus exposure. We do not have sufficient evidence to say whether the virus titer at 1 min of feeding is too low to detect in real-time PCR or *B. tabaci* needs a buffer time to direct its stylets to phloem tissue where the virions are abundant. The minimum phloem contact threshold period of around 1.8 min for successful inoculation of TYLCV by *B. tabaci* in tomato plants [46] justifies no detection of virus copies immediately after the feeding started. A maximum amount of 0.5–1.6 ng of TYLCV DNA was found in *B. tabaci* using quantitative chemiluminescent dot blot assay [16]. Similar dynamics of virus accumulation in *B. tabaci* were described by Kollenberg et al. [17]. The copies of bipartite watermelon chlorotic stunt virus (WmCSV) were 10-times higher than monopartite, TYLCV in *B. tabaci*. After feeding of 1 hr, 7.6E+4 copies of WmCSV genome were calculated in *B. tabaci*. Copies of WmCSV increased up to 4.7E+7 during first 16 hrs of feeding by *B. tabaci* [17]. In contrast, the present results indicate that copy number of monopartite, ChiLCV were 186-fold higher than bipartite, ToLCNDV in *B. tabaci* 1 hr post feeding. We estimated 4.92E+10 copies of ToLCNDV and 9.16E+12 copies of ChiLCV ingested by *B. tabaci* post 1 hr of feeding. The difference in copy number with previous studies may be due to variation in virus species and vectoring efficiency of *B. tabaci* cryptic species. Besides, it is important to note that virus concentrations in all the previous studies were measured in a group of *B. tabaci*, not in individual flies. The present study, for the first time, provided precise estimation of begomovirus copies by testing individual *B. tabaci* in real-time PCR.

Egestion of the virus copies by *B. tabaci* was recorded at different time intervals of feeding. Unlike virus acquisition, copy number of ToLCNDV was higher than ChiLCV in individual flies. The rate of egestion of both the viruses by *B. tabaci* followed a logarithmic model. ToLCNDV copy number was 10-fold higher than the ChiLCV at 6 hrs of feeding. The higher

copies of ToLCNDV indicates higher efficiency of *B. tabaci* Asia II 1 in releasing ToLCNDV as compared to ChiLCV. TYLCV, tomato yellow leaf curl China virus (TYLCCNV), tomato leaf curl Bangalore virus (ToLCBaV), papaya leaf curl China virus (PaLCuCNV), tomato leaf curl Taiwan virus (ToLCTWV), tomato yellow leaf curl Thailand virus (TYLCTHV), euphorbia yellow mosaic virus (EuMV), and tobacco curly shoot virus (TbCSV) are transmitted by different *B. tabaci* cryptic species at varying efficiencies [34, 40–42, 47–55]. A steady increase in the ToLCNDV copies was observed 6 hrs post feeding. However, we are not sure whether the estimation includes the replication of ToLCNDV in the detached leaves. The copies of ToLCNDV in detached leaf were higher than the copies acquired by *B. tabaci* in 24 hrs indicating replication of ToLCNDV in detached leaf. Replication initiator protein (Rep) and Replication enhancer proteins of TYLCV need to interact with host factor in order to create a cellular environment favorable for virus multiplication. It takes some time to recruit the other host factors and reprogramme the host cell cycle by Rep before the replication starts [56–58]. If this is the case, then the virus copies in the detached leaf during first few hours of feeding can be considered as the copies inoculated exclusively by *B. tabaci* which indicates around 10-fold higher egestion of ToLCNDV than ChiLCV. However, copies of ChiLCV in detached leaf 24 hrs post inoculation feeding was lesser than the copies *B. tabaci* acquired. Besides the low efficiency of *B. tabaci* Asia II 1 in inoculating ChiLCV, replication of ChiLCV in detached chilli leaf might also be sluggish than ToLCNDV in tomato leaf. Progressive displacement of tomato yellow leaf curl Sardinia virus by TYLCV due to higher transmission efficiency of TYLCV by *B. tabaci* MEAM1 came into light during tomato epidemics in Spain [59]. In Taiwan, MEAM1 was reported to transmit TYLCTHV virus more efficiently than ToLCTWV [54]. Coinfection of ToLCNDV and ChiLCV is often reported in solanaceous hosts. The synergistic or antagonistic interactions of ToLCNDV and ChiLCV will be worthy of future research to shed light on symptom expression, replication, and cell to cell movement of these two begomoviruses.

In conclusion, the present study, for first time, reports the copies of begomoviruses acquired and/or inoculated by individual *B. tabaci* and describes the comparative dynamics of ToLCNDV and ChiLCV. Moreover, the assay system developed in this study will be helpful for rapid screening of host resistance in large scale by quantifying virus copies in detached leaves by single whitefly inoculation.

## Supporting information

**S1 Table. Copies of ToLCNDV and ChiLCV ingested and egested by individual *B. tabaci* at different feeding exposure.**
(DOCX)

**S1 Fig.** Melt curves of ToLCNDV (a) and ChiLCV (b) amplicons in real-time PCR analysis indicated specificity of the reactions. The specific melting temperature for both ToLCNDV and ChiLCV products was around 81˚C. Standard curves show a linear relationship between log DNA concentrations in ng on X-axis and $C_T$ values on Y-axis for ToLCNDV (c) and ChiLCV (d). Each concentration was replicated thrice. The equation of the straight line and the coefficient of correlation ($R^2$) are mentioned on the graph.
(DOCX)

## Acknowledgments

The authors are thankful to Prof. Rajarshi Gaur, Prof Anupam Verma, Dr. T Makesh Kumar, Dr. Leonardo Velasco and one anonymous reviewer for editing, reviewing, and suggesting

substantial improvements in the manuscript. The authors acknowledge the infrastructural and manpower support received from IARI, New Delhi.

## Author Contributions

**Conceptualization:** Amalendu Ghosh.

**Formal analysis:** Buddhadeb Roy, Prosenjit Chakraborty.

**Investigation:** Buddhadeb Roy, Prosenjit Chakraborty.

**Methodology:** Amalendu Ghosh.

**Project administration:** Amalendu Ghosh.

**Resources:** Buddhadeb Roy, Amalendu Ghosh.

**Supervision:** Amalendu Ghosh.

**Writing – original draft:** Buddhadeb Roy, Prosenjit Chakraborty.

**Writing – review & editing:** Amalendu Ghosh.

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
