## [Decision Letter · Decision Letter 0]

10 Jun 2021

PONE-D-21-15811

How many begomovirus copies are acquired and inoculated by its vector, whitefly (Bemisia tabaci) during feeding?

PLOS ONE

Dear Dr. Ghosh,

Thank you for submitting your manuscript to PLOS ONE. After careful consideration, we feel that it has merit but does not fully meet PLOS ONE’s publication criteria as it currently stands. Therefore, we invite you to submit a revised version of the manuscript that addresses the points raised during the review process.

We look forward to receiving your revised manuscript.

Kind regards,

Rajarshi Gaur

Academic Editor

PLOS ONE

Journal Requirements:

3)  Thank you for stating the following in the Acknowledgments Section of your manuscript:

[The authors acknowledge the support received from IARI, New Delhi.]

 [The author(s) received no specific funding for this work]

Reviewers' comments:

Reviewer's Responses to Questions

**Comments to the Author**

1. Is the manuscript technically sound, and do the data support the conclusions?

Reviewer #1: Yes

Reviewer #2: Partly

Reviewer #3: Yes

2. Has the statistical analysis been performed appropriately and rigorously? 

Reviewer #1: No

Reviewer #2: N/A

Reviewer #3: Yes

3. Have the authors made all data underlying the findings in their manuscript fully available?

Reviewer #1: Yes

Reviewer #2: Yes

Reviewer #3: Yes

4. Is the manuscript presented in an intelligible fashion and written in standard English?

Reviewer #1: Yes

Reviewer #2: Yes

Reviewer #3: Yes

5. Review Comments to the Author

Reviewer #1: Roy et al. describes in this manuscript the quantification of the ingestion and egestion by Bemisia tabaci of two begomovirus species. The authors first identify the specific biotype of B. tabaci by PCR and sequence analysis of the mtCOI gene. They then design a experimental setup that allowed them to sample B. tabaci individuals that acquire begomoviruses, Tomato leaf curl New Delhi virus or Chilli leaf curl virus, by feeding on tomato and chili leaves, respectively. Virus egestion over time is also determined. Both viruses are quantified by qPCR in individual insects.

The authors determine that the dynamics of ingestion and egestion of the two viruses are differential. In the case of ToLCNDV, the dynamic curve of ToLCNDV ingestion follows a 3er order polynomial curve, which could indicate a saturation of the virus binding sites in the insect feeding apparatus. In the egestion of this virus and in the ingestion and egestion of ChLV, the dynamics obey a 2nd order polynomial equation.

The novelty of this study lies in the quantification of the viruses in individual insects and the identification of distinct dynamics of the ingestion and release of two viruses by B. tabaci.

In my opinion, these are interesting results that deserve publication, but first it is necessary to correct some aspects of the presentation and intrepretation of the results.

Minor comments

Figures 1 and 2 and Table 1 may go better in Supplementary material. Table 2 is graphically represented in Figure 3, so it is redundant, and is therefore either deleted or included in the Supplementary material.

Major comments

In Figure 3 the curves are not well distinguishable due to the type of representation. For a better visualization, the axis of the virus titers should be in logarithmic scale. I attach some figures of how they should be. Data points should be not connected with lines and should be plotted including the standard deviations of the means. On the other hand, any number of points can be connected to a polynomial equation, but this does not help its mechanistic interpretation. Since ingestion involves adsorption to a surface/protein (virion binding sites), it should be analyzed according to adsorption models such as the Langmuir, Freundlich or Langmuir-Freundlich isotherm models, just to cite some of the most common (e.g. see: 10.1016/j.jconhyd.2011.12.001). Once the type of isotherm that best fits the data (higher R2 value) has been obtained, the curve would be included in the same figure where the real data are represented. In this way, the ingestions (binding) and releases of both viruses could be interpreted mechanistically.

Reviewer #2: It is a good study to determine the number of virions acquired and transmitted by B.tabaci to infect plants with the two begomoviruses used in the study.

The MS needs minor revision to address some questions raised in the attached copy of the MS

Reviewer #3: It is a good piece of work which provided the information about the copy number acquired and egested by a single whitefly. Well laid out experiments brought out this authentic information and this findings support the earlier theory that a single whitefly can efficiently transmit the virus to a susceptible host which in turn can be a source of inoculam for fast spreading in the field.

Why there is a sudden fluctuation in copy number in case of ToLCNDV between 10 min to 6h of aquisition as well as in case of ChiLCV during 10 min to 12 hrs of aquisition?

whether virus propagation taken place with in the body and copy number increase occurred or not during the time point between aquistion and egestion. If information available on this may be added in the manuscript

6. PLOS authors have the option to publish the peer review history of their article (what does this mean?). If published, this will include your full peer review and any attached files.

Reviewer #1: No

Reviewer #2: **Yes: **Anupam Varma

Reviewer #3: **Yes: **Makeshkumar T

---

## [Author Response · Author response to Decision Letter 0]

6 Aug 2021

Response to reviewers’ comments:

We are thankful to the Reviewers and Academic Editor for critically going through the MS and suggesting substantial improvements. All the suggestions made by the reviewers have been incorporated into the revised MS. Our response to each comment is shown below. The changes made in the manuscript are indicated in red colour in the text. We trust that our revisions will make this MS acceptable for publication in the journal PLOS One.

Reviewer #1: Roy et al. describes in this manuscript the quantification of the ingestion and egestion by Bemisia tabaci of two begomovirus species. The authors first identify the specific biotype of B. tabaci by PCR and sequence analysis of the mtCOI gene. They then design a experimental setup that allowed them to sample B. tabaci individuals that acquire begomoviruses, Tomato leaf curl New Delhi virus or Chilli leaf curl virus, by feeding on tomato and chili leaves, respectively. Virus egestion over time is also determined. Both viruses are quantified by qPCR in individual insects.

The authors determine that the dynamics of ingestion and egestion of the two viruses are differential. In the case of ToLCNDV, the dynamic curve of ToLCNDV ingestion follows a 3er order polynomial curve, which could indicate a saturation of the virus binding sites in the insect feeding apparatus. In the egestion of this virus and in the ingestion and egestion of ChLV, the dynamics obey a 2nd order polynomial equation.

The novelty of this study lies in the quantification of the viruses in individual insects and the identification of distinct dynamics of the ingestion and release of two viruses by B. tabaci.

In my opinion, these are interesting results that deserve publication, but first it is necessary to correct some aspects of the presentation and interpretation of the results.

Minor comments

Comment 1: Figures 1 and 2 and Table 1 may go better in Supplementary material. Table 2 is graphically represented in Figure 3, so it is redundant, and is therefore either deleted or included in the Supplementary material.

Response: Figure 1 and 2 has been merged with Fig 3 as a single figure. Table 2 is graphically represented in Figure 3, so it has been revised as Supplementary material.

Table 1 contains primer sequences that have been designed and validated in this study. We prefer to keep this along with the main ms. 

Major comments

Comment 2: In Figure 3 the curves are not well distinguishable due to the type of representation. For a better visualization, the axis of the virus titers should be in logarithmic scale. I attach some figures of how they should be. Data points should be not connected with lines and should be plotted including the standard deviations of the means. On the other hand, any number of points can be connected to a polynomial equation, but this does not help its mechanistic interpretation. Since ingestion involves adsorption to a surface/protein (virion binding sites), it should be analyzed according to adsorption models such as the Langmuir, Freundlich or Langmuir-Freundlich isotherm models, just to cite some of the most common (e.g. see: 10.1016/j.jconhyd.2011.12.001). Once the type of isotherm that best fits the data (higher R2 value) has been obtained, the curve would be included in the same figure where the real data are represented. In this way, the ingestions (binding) and releases of both viruses could be interpreted mechanistically. 

Response: We agreed with the reviewer’s suggestion and have gone through the suggested literatures. However, the Langmuir, Freundlich or Langmuir-Freundlich isotherm models are adoptable for quantity of a gas adsorbed into a solid surface. In our case, we have virus quantity as one factor and another factor is time. Although the virus quantity adsorbed on the surface of the receptor, receptor surface area is not known in our study. Hence, the model could not be fitted in our data set. The time points of the present study could not be transformed in the model parameter. 

Reviewer #2: It is a good study to determine the number of virions acquired and transmitted by B.tabaci to infect plants with the two begomoviruses used in the study.

The MS needs minor revision to address some questions raised in the attached copy of the MS

Comment: Introduction needs to be rewritten to give more information about the transmission of Begomoviruses by whiteflies.

Response. Necessary information has been added as suggested. 

Comment: Please check and correct, Criniviruses, Ipomoviruses, and torradoviruses are shown to be transmitted by B. tabaci. 

Response: included in the revised ms. 

Comment: These references are not good here!!!

Response: Appropriate references have been cited in the revised ms. 

Comment: Check and use the correct number 

Response: revised as per latest ICTV checklist

Comment: Number not kinetics? Perhaps you want to say “The number of begomovirus particles acquired and transmitted by individual B. tabaci is important to shed light on the efficiency of virus transmission and useful in understanding the disease epidemiology.” Please check an correct. 

Response: Revised as suggested 

Comment: You have used similar conditions for the acquisition and inoculation of the viruses. Therefore the two paragraphs can be combined easily mentioning that for aquisition infected leaves for used and for inoculation healthy leaves were used

Response: merged as suggested 

Comment: How was the uniformity in size and weight of the leaves was maintained?

Response: Leaves of uniform shape and size were considered throughout the experiment. Mentioned in revised ms. 

Comment: Why not give actual calculated number instead of the formula as in the abstract you are talking about a million copies!!!

Response: This is not a formula. Because superscripted exponents like 107 cannot always be conveniently displayed, the letter E (or e) is often used to represent "times ten raised to the power of" (which would be written as "× 10n"). 

Comment: What was the difference in the number of copies in tomato and chili plants used for acquisition?

Response: The virus copies in source plants were almost equal. Now mentioned in revised ms. 

Comment: This needs explanation!!!

Response: included in discussion (line 282-291)

Comment: Did it match with the virus found in the whiteflies at different acquisition times?

Response: Virus copies increased with exposure in both acquisition and inoculation. However, the trend in increase was significantly different. Now, mentioned in discussion of revised ms. 

Comment: How did you compensate the multiplication of the viruses in the inoculated leaves?

Response: the multiplication of the virus in detached leaves cannot be compensate in the current methods. This has been mentioned in discussion (line 300-310). 

Comment: Did you test the leaves for virus titre after one minute inoculation feed at different time intervals to determine the minimum number of particles required for causing infection

Response: We agree with the reviewer’s suggestion. This would be an interesting study to understand the number of particles required to cause infection. However, this was beyond the scope of the current study. We will try to address this suggestion in our future studies. 

Comment: Change to ‘vectors’

Response: changed 

Comment: Change to a large number of viruses. There is no sanctity of the number 100!!! Even tomato is affected by nearly 100 species of begomoviruses.

Response: Changed

Comment: Authors should check the whole MS and be consistent. Compare this with the statement in the introduction.

Response: revised accordingly throughout the ms 

Comment: You may use virions at other places too, but make it uniform.

Response: mentioned as virions in all places as suggested 

Comment: Needs checking in the light of comments given earlier

Response: revised accordingly

Comment: Do you mean to say a million copies are required to cause infection?Please check and give convincing evidence.

Response: Revised as suggested

Comment: This is important, you need to check and confirm.

Response: Yes, we agree with the reviewers. Confirming the phloem contact time needs further in detail experiments and specialized instruments that is not feasible at this moment. 

Comment: This may vary from virus-host combinations. Just quoting one reference may not be correct. Moreover, this is not relevant here.

Response: deleted in revised version

Comment: Interesting this should have prompted testing at two minutes!!!

Response: This may vary with host-virus-vector combinations. In the present experiment, we have tested at 1 min and 5 min. 

Comment: Needs checking; see earlier comment. Please give actual calculated number

Response: It is the actual number. ‘E+10’ means 1010

Comment: Needs checking?

Response: The polynomial model has been removed as suggested by reviewer 1

Comment: Is it first time? What about WmCSV?

Response: The copy number of WmCSV was estimated in group of whitefly. The present study is the first to estimate the virus copies in individual whitefly

Comment: How?

Response: expanded in revised ms. 

Reviewer #3: 

It is a good piece of work which provided the information about the copy number acquired and egested by a single whitefly. Well laid out experiments brought out this authentic information and this findings support the earlier theory that a single whitefly can efficiently transmit the virus to a susceptible host which in turn can be a source of inoculam for fast spreading in the field.

Why there is a sudden fluctuation in copy number in case of ToLCNDV between 10 min to 6h of aquisition as well as in case of ChiLCV during 10 min to 12 hrs of aquisition? whether virus propagation taken place with in the body and copy number increase occurred or not during the time point between aquistion and egestion. If information available on this may be added in the manuscript

Response: This is not a sudden increase if you consider the duration of the virus exposure. The copies increased in individual whitefly with increase in acquisition time. The exposure to virus source for 6 to 12 hrs is the reason for increase in virus titre. 

Replication of TYLCV in whitefly has been reported. We have tested the same phenomenon for predominant begomovirus species in India in a different study. There was no evidence of replication of ToLCNDV and ChiLCV in whitefly.

---

## [Decision Letter · Decision Letter 1]

23 Sep 2021

PONE-D-21-15811R1How many begomovirus copies are acquired and inoculated by its vector, whitefly (Bemisia tabaci) during feeding?PLOS ONE

Dear Dr. Ghosh,

Thank you for submitting your manuscript to PLOS ONE. After careful consideration, we feel that it has merit but does not fully meet PLOS ONE’s publication criteria as it currently stands. Therefore, we invite you to submit a revised version of the manuscript that addresses the points raised during the review process.

We look forward to receiving your revised manuscript.

Kind regards,

Rajarshi Gaur

Academic Editor

PLOS ONE

Journal Requirements:

Reviewers' comments:

Reviewer's Responses to Questions

**Comments to the Author**

1. If the authors have adequately addressed your comments raised in a previous round of review and you feel that this manuscript is now acceptable for publication, you may indicate that here to bypass the “Comments to the Author” section, enter your conflict of interest statement in the “Confidential to Editor” section, and submit your "Accept" recommendation.

Reviewer #1: All comments have been addressed

Reviewer #4: All comments have been addressed

2. Is the manuscript technically sound, and do the data support the conclusions?

Reviewer #1: Yes

Reviewer #4: Yes

3. Has the statistical analysis been performed appropriately and rigorously? 

Reviewer #1: Yes

Reviewer #4: Yes

4. Have the authors made all data underlying the findings in their manuscript fully available?

Reviewer #1: Yes

Reviewer #4: Yes

5. Is the manuscript presented in an intelligible fashion and written in standard English?

Reviewer #1: Yes

Reviewer #4: Yes

6. Review Comments to the Author

Reviewer #1: Figure 1 has not been modified as suggested: Mean virus copies in the Y-axis should be in logarithmic scale for better visualization. Besides, figure 1c, 1e and 1f should be apart as e.g. figure 2, as they deal with different matters to those in figs. 1a and 1b. Nevertheless, I better think that Figs. 1c, 1e and 1f could be in Supplementary material.

To understand mechanistically the binding of viral particles and their egestion, the adsorption models outlined in my previous comments might still be useful, see Powell et al., 2000. Investigating the Effect of Carbon Shape on Virus Adsorption. https://doi.org/10.1021/es991097w. However, it is true that it is difficult to estimate the binding surfaces on the insect feeding apparatus, which might be too speculative. Therefore, I agree with the authors that their models of acquisition and egestion with time are good enough.

Reviewer #4: This study from Roy et al describes a novel method for quantification of acquisition & inoculation ToLCNDV and ChiLCV by Bemisia tabaci through detached leaf assay using real-time PCR. Overall, the experiments are well conceptualized and are of good scientific interest for future research on virus-vector epidemiology. The revised version of submitted manuscript has included major comments addressed from the three reviewers therefore, I recommend accepting the manuscript for publication. However, there are minor suggestions which should be incorporated in the final version.

1- The primers in Table-1 should be cross-checked as the sequence for AG153F and AG154R cannot be exact same therefore should be corrected. Also, the AG151F and AG153F are also similar, therefore please cross-confirm and apply corrections.

2- I would highly recommend the authors to provide a flow diagram or pictoral schematic template drawing for the entire protocol of ingestion and egestion by B. tabaci using detached leaf assay for better demonstration in the manuscript.

3-Please include error bars wherever applicable in the graphical data plots.

7. PLOS authors have the option to publish the peer review history of their article (what does this mean?). If published, this will include your full peer review and any attached files.

Reviewer #1: **Yes: **Leonardo Velasco

Reviewer #4: No

---

## [Author Response · Author response to Decision Letter 1]

2 Oct 2021

The authors are thankful to the Editor and reviewers for critically going htrough the manuscript and suggesting substantial improvement. All the suggestions made by the reviewers have been incorporated in the revised manuscript (R2). Our response to each comment is shown below. The changes made in the manuscript are indicated in red in the text. We trust that our revisions will make this MS acceptable for publication in Plos One. 

Reviewer #1: Figure 1 has not been modified as suggested: Mean virus copies in the Y-axis should be in logarithmic scale for better visualization. Besides, figure 1c, 1e and 1f should be apart as e.g. figure 2, as they deal with different matters to those in figs. 1a and 1b. Nevertheless, I better think that Figs. 1c, 1e and 1f could be in Supplementary material.

Response: Mean virus copies have been transformed in log scale as suggested. Fig. 1c-f have been revised as supplementary materials. 

To understand mechanistically the binding of viral particles and their egestion, the adsorption models outlined in my previous comments might still be useful, see Powell et al., 2000. Investigating the Effect of Carbon Shape on Virus Adsorption. https://doi.org/10.1021/es991097w. However, it is true that it is difficult to estimate the binding surfaces on the insect feeding apparatus, which might be too speculative. Therefore, I agree with the authors that their models of acquisition and egestion with time are good enough.

Response: Thank you. In accordance with the previous comment, the model has also been changed to a logarithmic model as the polynomial model did not fit in the revised figure after transformation of mean copy number into log scale. 

Reviewer #4: This study from Roy et al describes a novel method for quantification of acquisition & inoculation ToLCNDV and ChiLCV by Bemisia tabaci through detached leaf assay using real-time PCR. Overall, the experiments are well conceptualized and are of good scientific interest for future research on virus-vector epidemiology. The revised version of submitted manuscript has included major comments addressed from the three reviewers therefore, I recommend accepting the manuscript for publication. However, there are minor suggestions which should be incorporated in the final version.

1- The primers in Table-1 should be cross-checked as the sequence for AG153F and AG154R cannot be exact same therefore should be corrected. Also, the AG151F and AG153F are also similar, therefore please cross-confirm and apply corrections.

Response: Corrected in the revised ms. 

2- I would highly recommend the authors provide a flow diagram or pictoral schematic template drawing for the entire protocol of ingestion and egestion by B. tabaci using detached leaf assay for better demonstration in the manuscript.

Response: A pictorial diagram (Fig 1) has been provided as suggested. 

3-Please include error bars wherever applicable in the graphical data plots.

Response: The figure has been revised in light of the comments made by Reviewer 1. SEm values are included in Supplementary Table 1.

---

## [Editor Report · Decision Letter 2]

11 Oct 2021

How many begomovirus copies are acquired and inoculated by its vector, whitefly (Bemisia tabaci) during feeding?

PONE-D-21-15811R2

Dear Dr. Ghosh,

We’re pleased to inform you that your manuscript has been judged scientifically suitable for publication and will be formally accepted for publication once it meets all outstanding technical requirements.

Kind regards,

Rajarshi Gaur

Academic Editor

PLOS ONE